# CRYSTALS WITH TRANSFORMERS ON GRAPHS, FOR PREDICTIONS OF CRYSTAL MATERIAL PROPERTIES

## ABSTRACT

Graph neural networks (GNN) have found extensive applications across diverse domains, notably in the modeling molecules. Crystals differ from molecules in their ionic bonding across the lattice and ordered microscopic structures, which provide crystals unique symmetry and determine their macroscopic properties. Therefore, long-range orders are essential in predicting the physical and chemical properties of crystals. GNNs successfully model the local environment of atoms in crystals, however, they struggle to capture longer ranged interactions due to a limitation of depth. In this paper, we propose CrysToGraph (**Crys**tals with **T**ransformers **on Graph**s), a novel transformer-based geometric graph network designed specifically for crystalline systems. Notably, CrysToGraph effectively captures short-range dependencies with transformer-based graph convolution blocks as well as long-range dependencies with graph-wise transformer blocks. Moreover, it outperforms most existing methods, achieving new state-of-the-art results on the MatBench benchmark datasets.

## 1 INTRODUCTION

Graph Neural Networks (GNNs) represent a significant breakthrough in the field of machine learning when applied to graph-structured data. These networks are specifically designed to handle data organized as topological graphs. Such graph structures are prevalent in real-world scenarios, including knowledge graphs (Nathani et al., 2019; Hamaguchi et al., 2017; Schlichtkrull et al., 2018; Wang et al., 2018), social networks (Zhang and Chen, 2018; Qiu et al., 2018; Liu et al., 2019), recommendation systems (Ying et al., 2018; Monti et al., 2017; Berg et al., 2017), and also, natural science(Santoro et al., 2017; Battaglia et al., 2016; Duvenaud et al., 2015; Fout et al., 2017).

GNNs have also found great success in modeling small molecules (Kearnes et al., 2016; Xu et al., 2023; Cai et al., 2022; Zhu et al., 2022). Molecules, with atoms connected by covalent bonds, can be depicted as graphs naturally. This success with small molecules extends to related fields, such as inorganic crystals (Gong et al., 2022) and biological macromolecules (Zhang et al., 2022). Covalent bonds and Coulomb interactions are the primary forces responsible for packing atoms into crystals. GNNs are successful in capturing these short-range interactions. Yet, constrained by their depth, GNNs focus mainly on the local environment and struggle to capture global information in graphs. Hence, capturing information such as long-range orders which is crucial in crystalline systems is challenging for GNNs.

In traditional language models, like n-gram models (Cavnar et al., 1994) and recurrent neural networks (RNN) models (Rumelhart et al., 1985; Hochreiter and Schmidhuber, 1997) effectively model short-term dependencies but suffer long-term dependency problems. N-gram model is not capable for modeling long-term dependencies and RNN tends to lose distant contextual information. The emergence of transformer (Vaswani et al., 2017) addressed this issue by utilizing self-attention mechanism that updates the embeddings of each token based on the entire sequence. Inspired by the prosperity (Radford et al., 2018; Devlin et al., 2018) in natural language processing, we introduce transformer to capture long-range interaction information in crystals alongside GNN. In this paradigm, we aim to capture short-range interactions with the GNN and long-range order with a graph-wise transformer.

In this work, we present CrysToGraph, a transformer-based geometric graph network designed for crystalline systems. CrysToGraph employs a novel architecture that combines transformer-based

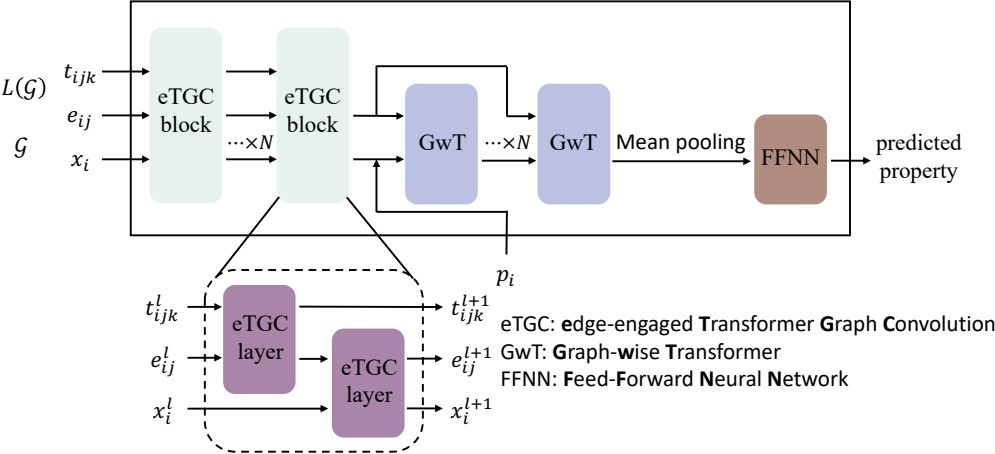

Figure 1: An overview of the architecture of CrysToGraph. In this paper, $\mathcal{G}$ denotes the original crystal graph, $L(\mathcal{G})$ denotes the line graph built upon the edges of direct crystal graphs. For inputs, $x_i$ denotes the atom (node) feature of node $i$, $e_{ij}$ represents the bond (edge) feature of edge $(i, j)$, $t_{ijk}$ represents the edge feature in the line graph, also the relationship between edge $(i, j)$ and $(i, k)$, $p_i$ denotes the positional encoding on atom $i$. Details of the graphs and positional encoding can be found in section 3.1 and 3.2. Details of the architecture can be found in section 3.3.

message passing blocks for updating node and edge features with a graph-wise transformer for explicitly incorporating long-range interactions. With line graphs for explicitly engaging the geometric and connectivity information, CrysToGraph outperforms most models on the MatBench benchmark datasets, achieving state-of-the-art results in a few datasets and establishing it as one of the best models for modeling crystal materials. We summarize our main contributions as follows:

1. We propose *CrysToGraph*, a transformer-based geometric graph network for explicitly capturing short-range and long-range interactions in crystalline systems, as shown in Figure 1.

2. We propose *eTGC* (edge-engaged transformer graph convolution), a transformer-based graph convolution layer that updates node features and edge features using a shared attention score calculated based on the features of the central node, neighboring nodes and edges.

3. We propose *GwT* (graph-wise transformer), a transformer encoder tailored for graphs, to capture long-range dependencies among nodes on a graph-wide scale.

4. We introduce spherical k-NN edge construction for building edge features in geometric graphs, effectively preserving connectivity and structure information for crystal structures.

## 2 RELATED WORKS

**Transformers on Graphs**  Originated from natural language processing tasks, transformer is designed to address the long-range dependency weakness in recurrent neural networks (RNN) based structures. Considering the great successes in natural language processing, transformer has been studied in graph domain, with expectations to help on the over-smoothing (Oono and Suzuki, 2019) and over-squashing (Alon and Yahav, 2020) problems in conventional GNNs. Dwivedi and Bresson (2020) firstly introduced transformer into graphs in Graph Transformer, with attention mechanism as a function of neighborhood connectivity. Eigenvectors of the Laplacian matrix were added to node features as positional encoding to strengthen the connectivity to the transformer-based message passing network in graph transformer. Hybrid architectures with global attention mechanism in addition to local message passing mechanism were introduced in GraphTrans (Wu et al., 2021) and GraphGPS (Rampášek et al., 2022). Kim et al. (2022) discarded local message passing structure and used standard transformers without graph-specific modifications in TokenGT and proved the capability with outstanding performance on large graph benchmarks. Pair-wise graph distances

were proposed in Graphormer (Shi et al., 2022; Ying et al., 2021), which achieved SOTA results on large-scale molecule benchmarks. Further research has been conducted in this field with SAN (Kreuzer et al., 2021), GraphiT (Mialon et al., 2021), SAT (Chen et al., 2022), EGT (Hussain et al., 2021), GRPE (Park et al., 2022), etc.

**Graph networks for crystal materials** In the context of chemical structures, which can be naturally represented as geometric graphs with nodes representing atoms and edges representing bonds, GNNs become instrumental in extracting structural and topological information and predicting physical and chemical properties. Although most GNNs work, CGCNN (Xie and Grossman, 2018) is the first one developed primarily focused on crystalline structures. CGCNN incorporates geometric construction of periodic multi-graphs and adopts a message-passing approach that concatenates node features from central and neighboring nodes, along with corresponding edge features. Subsequent models, like iCGCNN (Park and Wolverton, 2020) introduced Voronoi structures (Lee, 1982) for modeling three-body relation, GeoCGNN (Cheng et al., 2021) utilized attention masks and plane waves to encode local geometrical information, and MEGNet (Chen et al., 2019) incorporated global state information and edge updates. ALIGNN (Choudhary and DeCost, 2021) innovatively introduced line graphs to model geometric connectivity, while coGN and coNGN (Ruff et al., 2023) introduced nested graphs, which can be regarded as recursivly nested line graphs to explicitly model higher-ordered connectivity information. Matformer (Yan et al., 2022) firstly introduced attention mechanism with periodic pattern encodings into the modeling of crystalline systems. PotNet (Lin et al., 2023) attempted to incorporate inter-atomic potentials into the message passing framework to explicitly model the chemical interactions. Other techniques like contrastive learning (Bai et al., 2023; Kong et al., 2022) and prototypical classifiers (Bai et al., 2019) are also applied in the training of GNNs for crystals.

## 3 METHODS

### 3.1 CRYSTAL GRAPHS

The graphs are constructed from the structure of crystals. For every crystal, we construct the crystal graph with atoms as nodes. Edges between the nodes are identified in a k-nearest-neighbors manner. Line graphs are constructed based on the crystal graphs to explicitly model the connectivity and three-body interactions within the crystals.

#### 3.1.1 NODES

Node features play a pivotal role in the GNNs, and they represent the essential information pertaining to atoms within crystal structures. Each atom within the crystal is depicted as a node in the graph, with corresponding embeddings. The atom embeddings inherited the CGCNN atom embeddings of 92 dimensions. The CGCNN atom embedding is a curated set of features generated in one-hot encoding scheme. This encoding encompasses various atomic properties. These embeddings are atom-specific and do not inherently account for neighboring information or atomic charges.

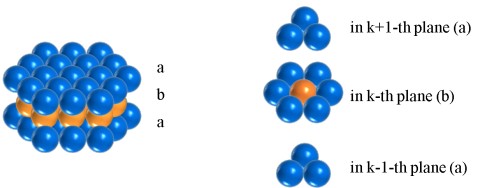

Figure 2: Left: A hexagonal close-packed (*hcp*) structure. Atoms stack by layers in an *abab...* pattern into a bulk crystal. Right: The theoretical maximum number of neighbors for a single atom (illustrated in orange) is 12.

#### 3.1.2 EDGES

Edges in crystal graphs represent the bonds connecting atoms, a fundamental aspect of inorganic crystal structures. The coordination number typically varies between 2 and 12 in crystal structures

(Wells, 2012). In the densest type of crystal structure, such as hexagonal close-packed ($hcp$) and cubic close-packed ($ccp$), each atom's coordination number reaches the theoretical maximum of 12, as shown in Figure 2.

In our study, we apply k nearest neighbor (k-NN) method to identify edges around nodes, setting the value of k to 12. This choice aligns with the theoretical maximum number of neighbors and aims to maximize the incorporation of neighbor information in each message-passing step. Importantly, it should be noted that these 12 nearest neighbors are considered connected, even if their physical distance may extend up to 20 Å.

To calculate edge features, we use the shifts in nodes' positions represented in spherical coordinates. These position shifts are expanded using radial-based filters to increase the dimension non-linearly. Additionally, we introduce a Boolean term in the edge features, indicating whether the distance between two neighbors exceeds the ion bond length cutoff. In this work, we define the threshold for a longest ion bond as 8 Å.

### 3.1.3 LINE GRAPHS

The line graph $L(\mathcal{G})$ of a given graph $\mathcal{G}$ is a graph where the nodes represent the edges in G, and the edges in $L(\mathcal{G})$ correspond to pairs of edges in $\mathcal{G}$, as illustrated in Figure 3. Specifically, for any pair of edges $(n_u, n_v)$ and $(n_v, n_w)$ in $\mathcal{G}$, the line graph $L(\mathcal{G})$ includes corresponding nodes $e_u$ and $e_v$. Moreover, there is an edge $(e_u, e_v)$ in $L(\mathcal{G})$, and the features of this edge are derived by expanding them using radial-based filters based on the cosine of the angle between the edges $(n_u, n_v)$ and $(n_v, n_w)$.

In essence, the line graph provides a higher-level representation where edges in the original graph become nodes, and connections in the line graph signify relationships between pairs of edges in the original graph, capturing information about their angles and structural configurations.

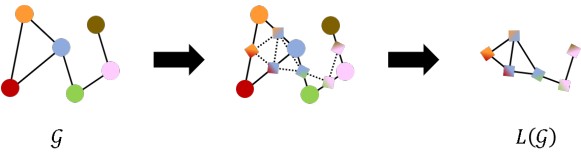

$\mathcal{G}$ $L(\mathcal{G})$

Figure 3: The construction of line graph $L(\mathcal{G})$ from a direct graph $\mathcal{G}$.

### 3.2 POSITIONAL ENCODING OF ATOMS

Conventional GNNs do not require positional encodings because they operate on graph structures that lack spatial information but inherently embody connectivity. However, in our work, we devise a graph-wise transformer structure to capture the long-range interactions, making positional encoding indispensable. Inspired by previous studies (Gao et al., 2022a;b), this structure incorporates positional encodings to effectively process spatial information.

To address this requirement, we have employed a comprehensive approach to positional encodings. We have combined multiple sources of positional information, including:

1. Cartesian coordinates: the absolute spatial positions of nodes in Cartesian coordinates.
2. Fractional coordinates: the relative position of nodes in the crystal cells.
3. Laplacian positional encoding (Dwivedi et al., 2020): a positional encoding based on the Laplacian operator, which captures structural relationships within the graph.
4. Random walk positional encoding (Dwivedi et al., 2021): a positional encoding derived from random walk processes, providing additional information about connectivity.

By concatenating these positional encodings, we aim to provide a holistic representation of spatial information within the graph. This comprehensive approach ensures that the positional encodings capture both the absolute and relational aspects of node positions and the connectivity, facilitating the effectiveness of the subsequent graph-wise transformers in our model.

### 3.3 ARCHITECTURE

We here introduce the architecture of CrysToGraph. The model contains 3 parts: edge-engaged transformer graph convolution for modeling short-range interactions, graph-wise transformers for modeling long-range interactions and feed forward linear layers for predicting of task-specific properties. The input crystal graphs consist of the direct graphs and line graphs, the outputs of the entire model are the properties of the crystals. The detailed structure is shown in Figure 4, large version can be found in appendices.

The CrysToGraph model can be denoted as:

$$CrysToGraph(\mathcal{G}, L(\mathcal{G})) = FFNN(GwT_{\times N}(eTGC_{\times N}(\mathcal{G}, L(\mathcal{G}))))$$

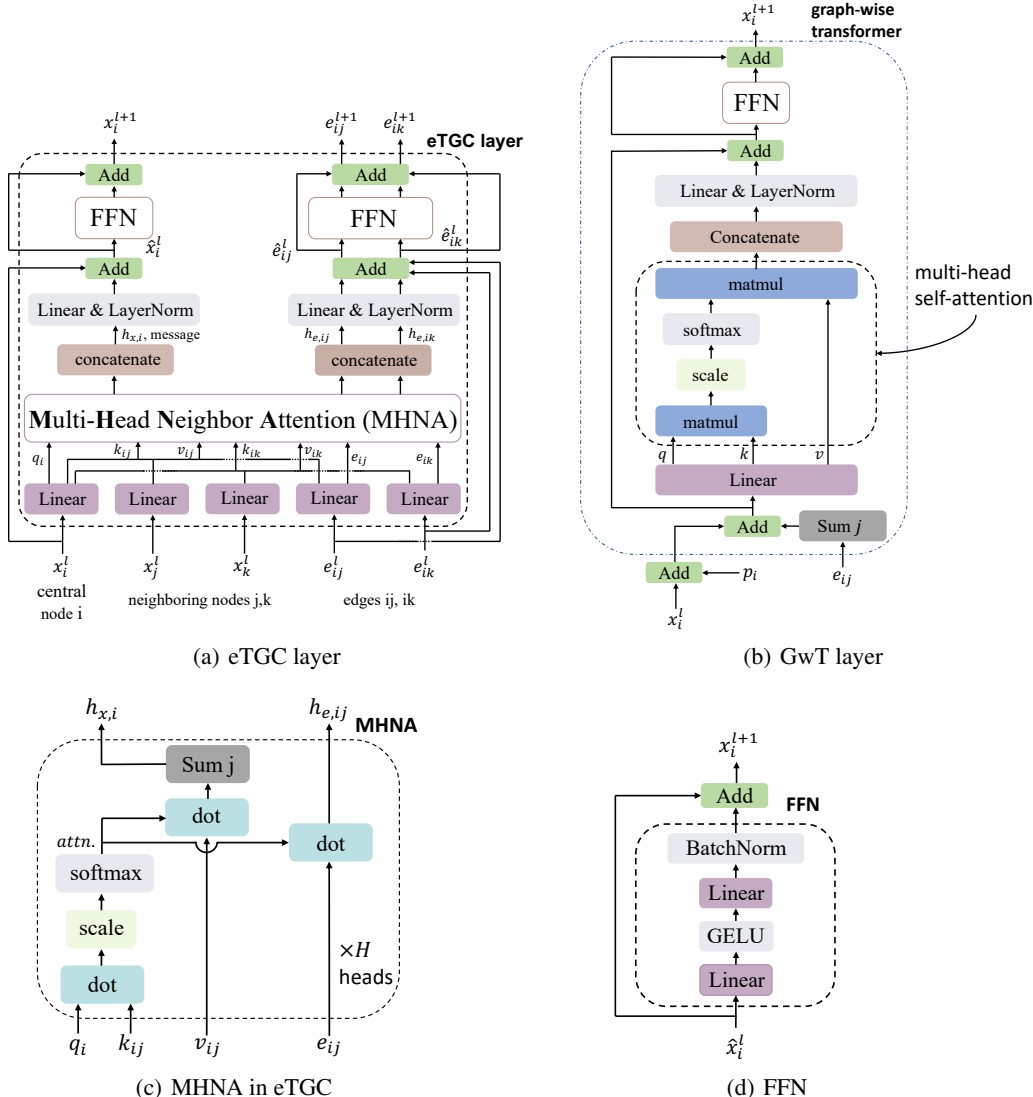

Figure 4: Structure of (a) an eTGC layer, (b) a GwT layer, (c) multi-head neighbor attention in eTGC layers, (d) FFN in eTGC layers and GwT layers.

### 3.3.1 EDGE-ENGAGED TRANSFORMER GRAPH CONVOLUTION (ETGC)

In each eTGC blocks, of which consist two eTGC layers for direct graphs and line graphs respectively, node features, edge features and edge features of the line graphs are updated. The eTGC layer

for line graphs updates the edge features of direct graphs and the edge features of the line graphs first, then the eTGC layer of direct graphs updates the node features and edge features of direct graphs. An eTGC layer contains a linear transformation, multi-head neighbor-attention (MHNA) and a feed-forward network (FFN).

In each eTGC layer, input node features are linear transformed into query, key and value vectors $Q$, $K$ and $V$. The edge features are linear transformed into $E'$:

$$Q = W_q X, K = W_k X, V = W_v X, E' = W_e E$$

Scoping to a single central node and its neighbors, $k$ and $v$ of both central node $i$ and neighboring nodes $j$ and $k$ are concatenated with edge feature $e'$ in a CGCNN manner. The neighbor-attention is calculated with $q_i$ of central node and linear transformation results of the concatenated vectors $k_{ij}$ and $v_{ij}$. Node features and edge features are updated with shared attention scores which are scaled with constant $\sqrt{d_k}$ and softmax layer. For simplicity of illustration, we consider a single-head attention and assume $d_k = d_v$:

$$k_{ij} = W_{ke}(k_i, k_j, e'_{ij}), v_{ij} = W_{ve}(v_i, v_j, e'_{ij})$$
$$attn = softmax(\frac{q_i k_{ij}}{\sqrt{d_k}})$$
$$h_{x,i} = \sum attn \cdot v_{ij}, h_{e,ij} = attn \cdot e'_{ij}$$

where $h_x$ and $h_e$ denotes the hidden outputs of the multi-head neighbor-attention. Outputs of all heads are concatenated and linear transformed into the hidden outputs, among which the hidden output $h_{x,i}$ on the certain node $i$ is the message being passed to $i$.

Scoping back to the graph, hidden outputs of the multi-head neighbor-attention are linear transformed and layer normalized before adding to the input central node features or edge features:

$$\hat{x}^l = x^l + LayerNorm(W_{on}{h_x}^l)$$
$$\hat{e_{ij}}^l = e_{ij}^l + LayerNorm(W_{oe}{h_e}^l)$$

where $\hat{x}$ and $\hat{e}$ denote the node outputs and edge outputs of the multi-head neighbor-attention at each node or edge.

Each multi-head neighbor-attention is followed by a feed-foward network in the eTGC layer. The FFN contains a batch normalization and two linear layers with GELU as activation function in the middle. A residual connection is applied at the end:

$$x^{l+1} = \hat{x}^l + BatchNorm(W_{x2}GELU(W_{x1}\hat{x}^l + b_{x1}) + b_{x2})$$
$$e^{l+1} = \hat{e}^l + BatchNorm(W_{e2}GELU(W_{e1}\hat{e}^l + b_{e1}) + b_{e2})$$

where $GELU(x) = xP(X < x) = 0.5x(1 + erf(\frac{x}{\sqrt(2}))$.

### 3.3.2 GRAPH-WISE TRANSFORMER (GwT)

Chemically, long-range order distinguishes crystals from small molecules by providing them with a regular structure, macroscopic symmetry, and unique optical properties, while short-range interactions are responsible for stabilizing the crystal structure on a local scale. While the eTGC blocks focus on the short-range interactions in local scale, we implemented GwT to explicitly model the long-range interactions within the entire crystal.

In this GwT, no connectivity information is explicitly incorporated since all nodes in each graph are treated as an ordered sequence of tokens. Thus, positional encodings and the sum of incoming edge features are added to the node features in prior:

$$x_{i,pe} = x_i + W_{pe}p_i + \sum_j W_e e_{ij}$$

where $p_i$ denotes the positional encoding of node $i$.

Typical multi-head self-attentions are applied on node features in the entire graphs, followed by a residual feed forward network with layer normalization:

$$Q = W_q X, K = W_k X, V = W_v X,$$
$$Attn = softmax(\frac{QK^T}{\sqrt{d_k}}),$$
$$H^l = X^l + LayerNorm(W_o(Attn \cdot V^l)),$$
$$X^{l+1} = H^l + LayerNorm(W_2 GELU(W_1 H^l + b_1) + b_2)$$

### 3.3.3 Feed-Forward Neural Network (FFNN)

To capture both short-range and long-range interactions within the crystals, we employed eTGC blocks and GwT. These components model the interactions at different scales. Each graph's node features are aggregated by taking their mean, resulting in a graph-level feature. This feature is then layer-normalized and fed into a task-specific feed-forward neural network for predicting specific properties. The task-specific feed-forward neural network we implemented is a multi-layer perceptron, utilizing softplus as activation functions:

$$X^l = Softplus(W_l X^{l-1} + b_l),$$
$$X^{out} = W_n X^{n-1} + b_n$$

where $X^{out}$ denotes the final prediction if the task is a regression task. In classification tasks, the positive $X^{out}$ yields positive predictions, negative $X^{out}$ yields negative predictions.

## 4 Experiments

### 4.1 Datasets

The experiments are conducted on the Materials Project database (Jain et al., 2013). We explore the general structure of the CrysToGraph on two datasets, namely `mp_e_form` and `log_gvrh`, with 132,752 and 10,987 samples. The average atoms in a single cell in the two datasets are 29.1 and 8.6, respectively. Results on the two datasets fairly demonstrate the performance on datasets with large and small cells. We explored the function of each part and optimized the hyperparameters. Here we focus on the depth and how the two compositions co-orperate. More training details can be found in appendices.

We evaluated the performance on MatBench benchmark (Dunn et al., 2020), a compilation of 13 datasets, providing structures in some datasets and composition in others. The compositions do not contain any structural information while crystal structures do. We trained and evaluated our model on 8 datasets with structures as the raw data. As crystal graphs are constructed based on the crystal structure, we trained and evaluated our model only on those datasets with structures as input. The targets of the datasets include exfoliate energy, formation energy, dielectric constant, etc. The dataset sizes span a wide range, ranging from 636 to 132,752 crystal structures. The task `mp_is_metal` is binary classification, while the rest are regression.

### 4.2 Exploration of Network Structure

**eTGC** The eTGC layer, as a message passing block, is designed to capture short-range dependencies. eTGC layers contribute to a majority in the overall performance. Networks with only eTGC blocks can even outperform networks with both eTGC and GwT blocks on datasets with smaller graphs when eTGC is deep enough, although a shallower eTGC combined with GwT layers still outperforms others.

**GwT** The GwT layers primarily designed for modeling long-range interactions can also capture short-range interactions, to a limited extent. Networks built only with GwT layers and task-specific feed-forward neural network fail to fully model all interactions within the graph, however, we can still find some insights from this set of experiments.

Figure 5 shows that eTGC contributes the most part to the overall performance since covalent bond and ionic bond in short range are the major chemical interactions in the crystal. However, the existence of GwT improves the ability to capture all interactions within the crystals. If eTGC is absent, a shallow GwT of 1 layer does not capture as much information as a 3-layered or deeper

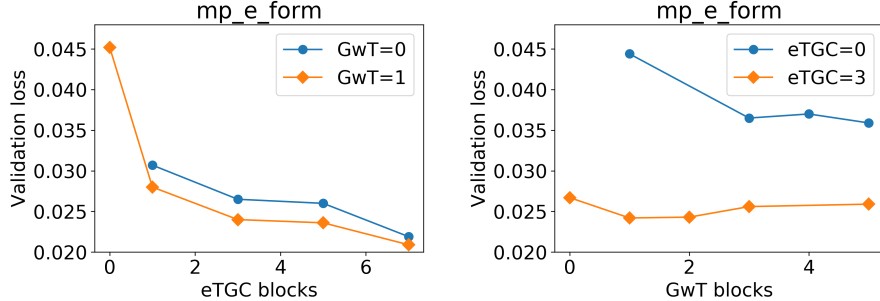

Figure 5: Left: Various depth of eTGC with fixed depth of GwT on `mp_e_form` dataset. Right: Various depth of GwT with fixed depth of eTGC on `mp_e_form` dataset.

GwT. However, an additional 1 layer of GwT is usually enough for capturing long-range interactions when the short-range interactions are modeled by eTGC.

It is worth noting that, if GwT follows eTGC blocks, deeper GwT negatively affects the overall performance, while 1 layer strikes the best score. This can be explained by GwT's preference in modeling global environment rather than the local ones. Too deep GwT tends to draw too much attention to the macroscopic symmetry and long-range interactions, forgetting the local chemical environment modeled by eTGC.

Comparing the left of Figure 5 and the left of Figure 6, the CrysToGraph performs differently on dataset `mp_e_form` with large cells of 29.1 atoms in average and dataset `log_gvrh` with small cells of 8.6 atoms in average. On small cells, a 5-block eTGC is enough for capturing most interactions in the graphs. However, on large cells, GwT is essential in capturing long-range interactions and overall information, since infinite depth of eTGC is not practical.

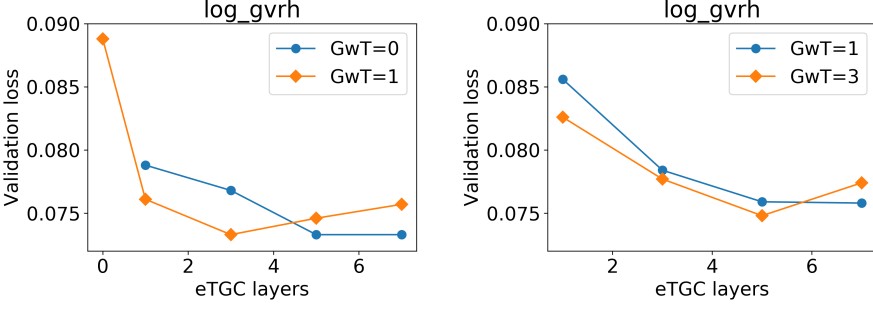

Figure 6: Left: Various depth of eTGC with fixed depth of GwT on `log_gvrh` dataset. Right: eTGC and GwT co-operate in parallel and the performances on `log_gvrh` dataset

**Merging eTGC and GwT** The relationship between the two major compositions is crucial to the successful modeling of crystals. As illustrated in Figure 7, we have three hypotheses about the proper structure for modeling short-range and long-range chemical interactions: stacking in sequence with eTGC first, stacking with GwT first and running in parallel. We tested these hypotheses with fixed depth of eTGC and GwT.

The right of Figure 6 demonstrates the performance trend with a parallel structure. Unlike the behavior with a stacking structure, deeper GwT tends to yield lower validation loss. However, if eTGC is sufficiently deep, deeper GwTs are still not favored.

The optimal depth of GwT depends on how the two components assemble. With a parallel structure, deeper GwT blocks tend to yield better results, except with excessively deep eTGC blocks. However,

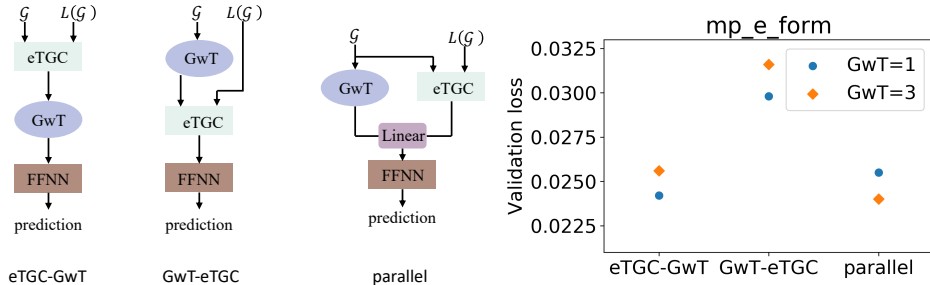

Figure 7: Left: Illustration of the three types of relationship between 3-layered eTGC and GwT blocks. Right: Performances on `mp_e_form` dataset with three types of relationship.

Table 1: Results on MatBench benchmark

| Datasets | CrysToGraph | coGN | coNGN | ALIGNN | CGCNN | MODNet | Matformer |
|---|---|---|---|---|---|---|---|
| dielectric | 0.3084 | 0.3088 | 0.3142 | 0.3449 | 0.5988 | **0.2711** | 0.634 |
| jdft2d | **32.3720** | 37.1652 | 36.1698 | 43.4244 | 49.2440 | 33.1918 | 42.827 |
| log_gvrh | 0 .0731 | 0.0698 | **0.0670** | 0.0715 | 0.0895 | 0.0731 | 0.077 |
| log_kvrh | 0.0543 | 0.0535 | **0.0491** | 0.0568 | 0.0712 | 0.0548 | 0.063 |
| mp_e_form | 0.0191 | **0.0170** | 0.0178 | 0.0215 | 0.0337 | 0.0448 | 0.0212 |
| mp_gap | 0.1714 | **0.1559** | 0.1697 | 0.1861 | 0.2972 | 0.2199 | 0.1878 |
| mp_is_metal | 0.9146 | 0.9124 | 0.9089 | 0.9128 | **0.9520** | 0.9038 | 0.906 |
| phonons | **28.3990** | 29.7117 | 28.8874 | 29.5385 | 57.7635 | 38.7524 | 42.526 |

this does not indicate a better performance with eTGC and GwT in parallel than those with stacking structure. In a stacking structure, eTGCs followed by 1 layer of GwT outperforms all other models.

### 4.3 RESULTS ON BENCHMARKS

Detailed result is in Table 1. CrysToGraph achieved the state-of-the-art in the `phonons` and `jdft2d` dataset. In other tasks, although not the top one, our model ranks among the top three in 5 datasets out of 6.

Results of current benchmark holders are also listed for comparison, among which Matformer was re-trained by Ruff et al. (2023). Mean absolute error (MAE) and area under the receiver operating characteristic curve (AUC) are calculated as the metrics to keep aligned with the official benchmark.

## 5 CONCLUSION

In this paper, we propose CrysToGraph, a geometric graph network that captures short-range and long-range interactions with transformer-based message-passing blocks (eTGC) and graph-wise transformers (GwT). We discuss the roles of the two components and how they assemble. We conclude that the eTGC blocks mainly function as capturing short-range interactions and GwT layers mainly function as capturing long-range interactions. We evaluate our model on MatBench benchmark, on which our model achieves state-of-the-art results.

Although our model achieves promising results on MatBench benchmark, we do believe there is still some space for our model to improve. We did not further optimize due to a limitation in computation resource. Future work may also modify our model to E(3)-invariant GNNs and fit for further applications on machine learning force field predictions.

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

# Appendices

## 1 ARCHITECTURE OF CRYSTOGRAPH

Figures in this section are same as Fugure 4 in the main text. Limited by space in the main text, we keep the figures in the main text compact and leave the large version here in Figure A1 - A2.

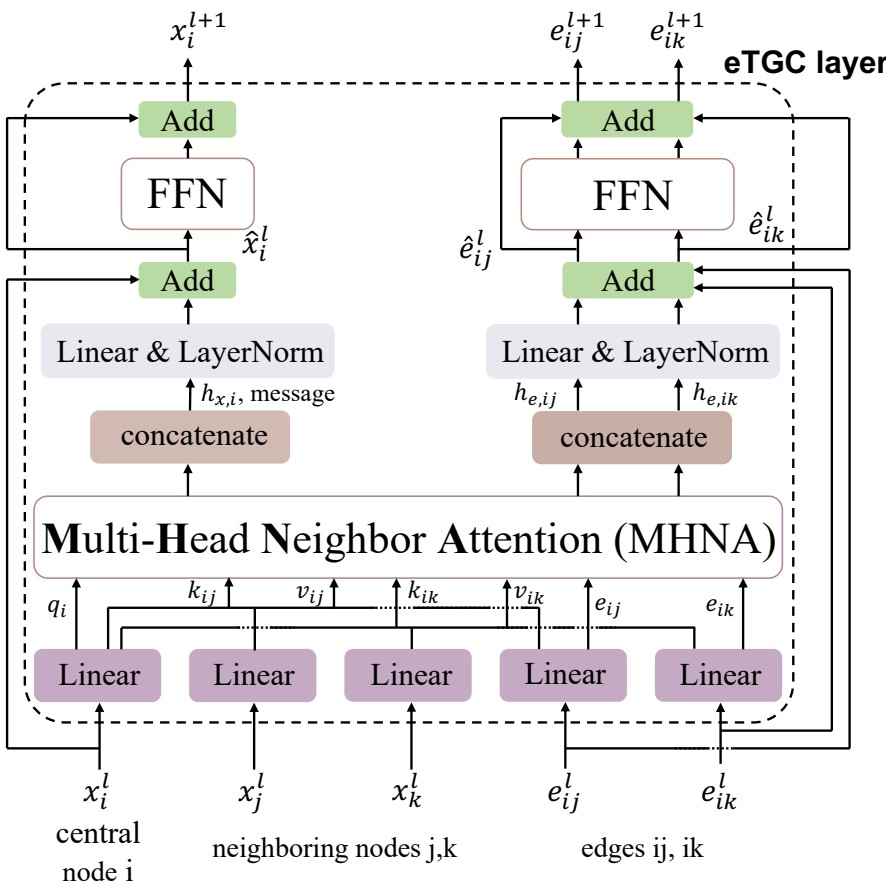

Figure A1: Structure of an eTGC layer.

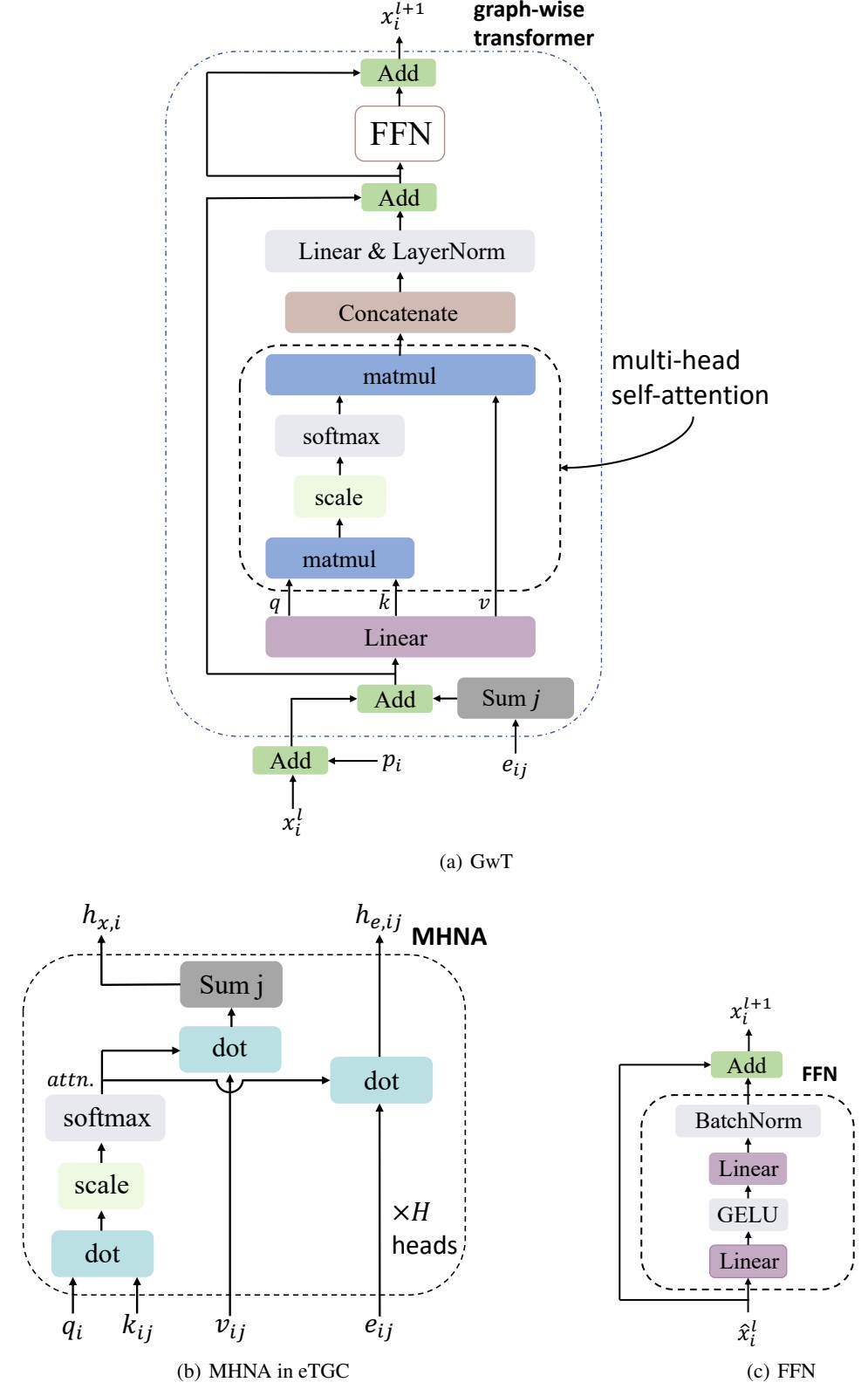

Figure A2: Structure of (a) a GwT layer, (b) multi-head neighbor attention in eTGC layers, (c) FFN in eTGC layers and GwT layers.

## 2 MASKED ATOM PRETRAINING

In the pretraining phase for atom representations, we introduced a masked atom prediction task. In this task, a specified percentage of atoms within each graph are masked. Specifically, 15% of the atoms in each graph are subjected to masking operations. Among these, 80% are substituted with a designated mask token, while 10% are replaced with randomly selected tokens, and the remaining 10% are left unchanged. In instances where the number of nodes in a graph is insufficient to maintain the masking rate below 15%, the crystal structure is expanded in all three dimensions. With graphs constructed as aforementioned, we trained a CGCNN model on these constructed graphs to predict the types of masked atoms. The loss curve is in Figure A3.

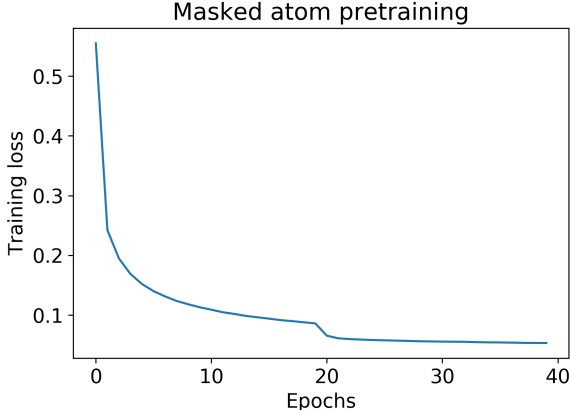

Figure A3: The loss curve in atom masked pretraining. The learning rate decrease by 10 at epoch 20.

Following the pretraining of atom embeddings, we concatenated the machine-learnt embeddings with the manually curated CGCNN atom embeddings.

## 3 LAYERS OF FFNN

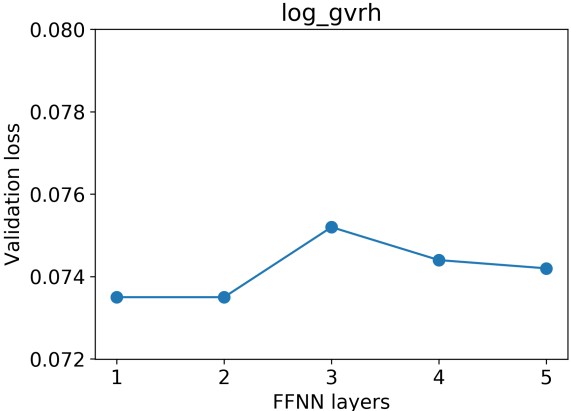

Figure A4: Various depth of FFNN, trained on `log_gvrh` dataset.

We can see in Figure A4, deeper FFNN impedes the overall performance. The ideal depth of FFNN is 1 or 2 layer.

# 4 LEARNING RATE

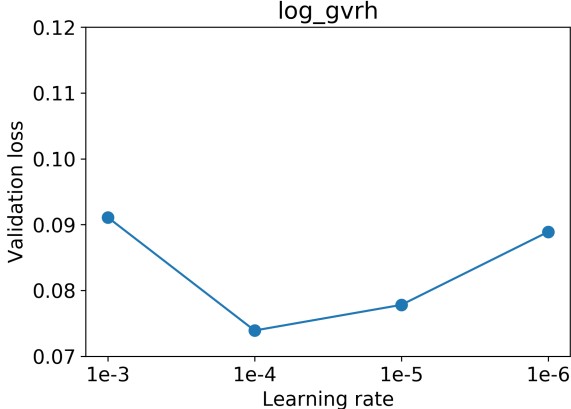

Figure A5: Same model trained in various learning rate.

Shown in Figure A5, the optimal learning rate is 1e-4, which is generalized to other experiments in this work.

# 5 WEIGHT DECAY

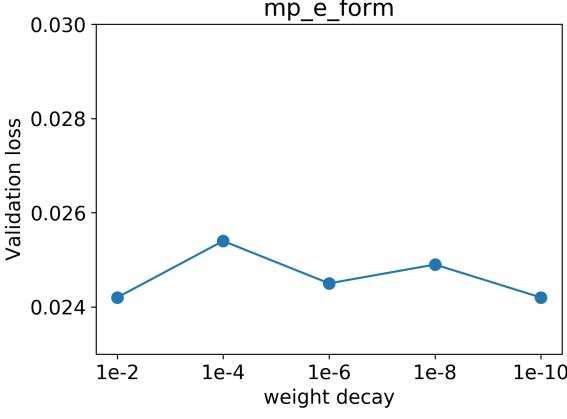

Figure A6: Models trained with various weigh decay penalty.

The weight decay can be regarded as a derivative of L2 regularization. Our model is excessive in parameter, however, the performance does not change much when the weight decay penalty varies, as shown in Figure A6.

# 6 ABLATION STUDIES ON POSITIONAL ENCODING

The positional encoding for the graph is composed of four parts: Cartesian coordinate, fractional coordinate, Laplacian positional encoding and random walk positional encoding. As shown in Figure A7 ,the Laplacian positional encoding and random walk positional encoding contribute the major part in encoding the connectivity and structure of the graphs.

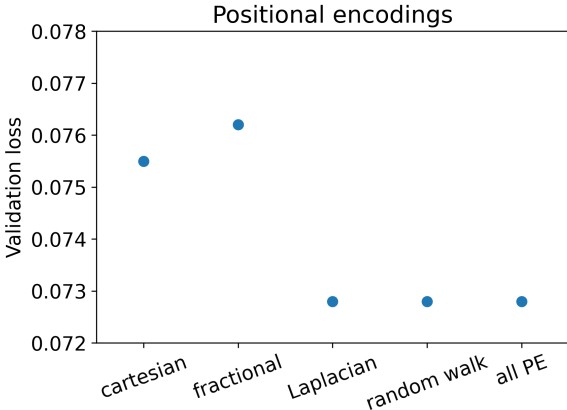

Figure A7: Ablation studies on positional encoding before the GwT.

Table A1: Results on MatBench benchmark

| Datasets | Targets |
|---|---|
| dielectric | Refractive index (unitless) |
| jdft2d | Exfoliation energy ($meV/atom$) |
| log_gvrh | Base 10 logarithm of the DFT Voigt-Reuss-Hill average shear moduli in GPa |
| log_kvrh | Base 10 logarithm of the DFT Voigt-Reuss-Hill average bulk moduli in GPa |
| mp_e_form | Formation energy ($eV/atom$) |
| mp_gap | Band gap ($eV$) |
| mp_is_metal | Binary, 1 if structure is metal otherwise 0 |
| phonons | Frequency of the highest frequency optical phonon mode peak ($cm^{-1}$) |

## 7  DETAILS OF DATASETS

We evaluated our model on 8 datasets from the MatBench benchmark. The targets include a wide range from microscopic properties like energy to macroscopic properties like refractive index. The target of tasks are shown in Table A1.

