# OpenReview forum: "Crystals with Transformers on Graphs, for predictions of crystal material properties"
_ICLR.cc/2024/Conference — Submitted to ICLR 2024_

### Official Review · Reviewer_5cNz · 2023-10-16

**Soundness:** 4 excellent
**Presentation:** 4 excellent
**Contribution:** 3 good
**Rating:** 6
**Confidence:** 3

**Summary:**

This paper proposes a general framework, CrysToGraph, for crystal materials property prediction. The overall idea is to encode the crystal graph into an embedding representation, and use this representation for classification or regression tasks. More specifically, the inputs are composed of the atom node embeddings, edge embeddings and angle information for edge pairs. These features first go through edge-engaged transformer graph convolution (eTGC) to extract phase-1 embeddings for atoms and edges. Then the Graph-wise Transformer (GwT) iteratively update atom node encodings by absorbing `information` from connecting edges. The final output goes through an FFN layer and produce an embedding for regression or classification task. The proposed model is tested on 8 datasets and achieves significant improvements over prior papers.

**Strengths:**

1. Crystal materials property prediction is a complicated natural science task. I think the authors did a good job on delineating and explaining the concepts of this task, and break down break down into smaller modules easier to understand.
2. In section 3.3, while I think the contents are organized well by explaining the architecture of each sub-module one by one, it would be better to further explain the motivations behind each sub-module, since the proposed system is large and complicated. I can see section 4.2 gives some hints, but some more elaborated explanations would be better.
3. The authors did a quite thorough analysis on model depth and other hyper-params, on up to 8 datasets, comparing against 6 baselines. The results look promising and convincing.

**Weaknesses:**

1. In section 3.1.3, if I understand correctly, $u'$ and $v'$ actually represent edges? If so, it's better to use $e$ notations to differ from the atom nodes. If this purely represents the angle between edges, pls make this definition more clear and draw connections with $t_{ijk}$ from Fig. 1.
2. In section 3.2, I understand the motivations of adding positional encoding. But I hope authors can give more insights on why multiple encodings need to be used? What if you only choose 1 or 2? Some ablation studies on this point would be helpful.
3. There are many existing works on crystal materials property predictions. For example, [1, 2] uses GNN and contrastive learning to predict density of states, and [2] uses prototypical classifiers to analyze crystal structures. These works are worth discussing in the related work.
4. To facilitate understanding, it would be better if you can use a running example (e.g., one simple crystal structure) like Fig 2 to explain how each module and inputs/outputs are hooked.

[1] Bai, J., et al. "Xtal2DoS: Attention-based Crystal to Sequence Learning for Density of States Prediction." NeurIPS 2022 AI for Science: Progress and Promises. 2022.

[2] Kong, S., et al. 2022. Density of states prediction for materials discovery via contrastive learning from probabilistic embeddings. Nature communications, 13(1), p.949.

[3] Bai, J., et al. "Imitation refinement for x-ray diffraction signal processing." In ICASSP 2019.

**Questions:**

1. I understand you used kNN to build up edges between atoms. I wonder what if you choose k=8 or k=6. How much difference does it make? Also, is it for simplicity purpose that you set the same k for all atom nodes?
2. For GwT, it seems that only the neighboring edges are added through $e_{ij}$, why it can capture long-range interactions? Do you expect that when N grows larger, it keeps drawing information from more distant neighbors?
3. How is the speed when training such a model? Are all the sub-modules trained together? Is there any speed analysis?
4. I didn't see $t_{ijk}$ in Fig 4. Is it used in encoding?
5. For edge-engaged transformer graph convolution, does it involve any convolution operations in Fig. 4 (a)?

---

> ### Author Response · Authors · 2023-11-22
>
> Dear reviewer,
>
> Thank you for taking the time to share your thoughts and experiences with us. Your insightful feedback is valuable and has provided me with an opportunity to delve deeper into the nuances of our work. Upon careful consideration of your review, I would like to address some of the concerns and criticisms raised. Firstly, I appreciate the attention drawn to the areas where the manuscript may benefit from further elaboration or clarification. I am committed to revising the paper to provide a more comprehensive and lucid explanation of the underlying concepts.
>
> I express my heartfelt appreciation for your recognition of the significance of my contributions in the field of computational materials research as well as the constructive advices. For the weakness you pointed out, we took this matter in the highest regard and would like to response and clarify as follow:
>
> 1.	$\displaystyle u’ $ and $\displaystyle v’ $ represent edges, while $\displaystyle (u’, v’) $ represents $\displaystyle t_{ijk} $. I will use e notations as you advised in the revised edition.
> 2.	I conducted an ablation study on the positional encoding, but I did not include this in my paper. PE 1 and 2 are the encoding for the structural information while PE 3 and 4 encode the connectivity, neither of them are neglectable. The ablation study proved this as well: the result in MAE has such order: with all PE < with PE 3 or 4 << with PE 1 or 2 < without PE.
> 3.	There are many existing works on this topic, where I may have missed some important ones. I will read these papers and add them to the discussion in the revised edition.
> 4.	That is a good idea to make the paper clearer and easier to understand. I will try to add this in the revised edition.
>
> For the questions, I would answer as follow:
> 1.	I have also tried k=8 and varied k. Firstly, k=8 and k=6 are less reasonable chemically than k=12 because there are many cases in the crystals that atoms have 12 neighbors such as close-packed structures (see the right half of Figure 2, 12 atoms surround the orange atom in the center). This is also proved in the test result: with the same split in training set and validation set, the MAEs are 0.0242, 0.0261, 0.0346 for k=12, varied k, k=8, respectively. The varied k may lead to some loss of structural information of local environment and too much complexity.
> 2.	The GwT is a graph-wise transformer, capturing long-range interactions by calculating the attention across all nodes in the graph, with structural and connectivity information added as the PE. The neighboring edges are also considered as a supplementary of the PE for connectivity.
> 3.	The speed was normal, slightly slower than ALIGNN which also take direct graph $\displaystyle \g G $ and line graph $\displaystyle L(\g G) $ as input. All sub-modules are trained together. We tried a contrastive pretraining with some data augmentations and masking, however, it did not significantly improve the performance. We did not make a full analysis on speed.
> 4.	It is not used in encoding. We have 2 eTGC layers in a single eTGC block, taking $\displaystyle e $, $\displaystyle t $ and $\displaystyle x $, $\displaystyle e $ as input respectively. tijk is the edge feature in the line graphs, representing the angle between the bonds. We only demonstrated one eTGC layer in Figure 4 for simplicity, while there are 2 eTGC layers taking different input.
> 5.	The message passing with transformation is mainly in the multi-head neighbor attention, which was expanded in Figure 4c.
>
> Best Regards,
>
> Authors of CrysToGraph

---

### Official Review · Reviewer_uVzi · 2023-10-26

**Soundness:** 3 good
**Presentation:** 2 fair
**Contribution:** 2 fair
**Rating:** 3
**Confidence:** 2

**Summary:**

Paper aims to use a graph-transform to predict properties of crystal-graphs. Notably, transformers are well suited for this task because unlike traditional MPNNs they are able to capture both local and global information.

The paper has some interesting ideas, but it appears to be mostly a mixing and matching of existing methods with only moderately impressive numerical experiments. If it were more clearly different than existing methods or if it was clear that this method of combining existing results lead to a massive improvement numerically, I would more favorably inclined. However, as is, I do not think this paper is good enough in its current form.

That said, I am not an expert on transformers and have no experience with crystals. Therefore, my opinion should be taken with a grain of salt.

**Strengths:**

Numerical results seem moderately impressive and the authors seem knowledgeable about GNNs / Transformers in the context of crystallography.

**Weaknesses:**

Method seems only slightly different than existing approaches (essentially being a linear combination of existing methods) and only achieves moderately good numerical results.

Less importantly, there are also numerous typos and grammar / spelling errors.

**Questions:**

What is meant by coordination number?

Is the $k$-nn graph an "or" k-nn, and "and" k-nn, or a directed graph?

---

> ### Author Response · Authors · 2023-11-22
>
> Dear reviewer,
>
> Thank you for your time on reviewing our manuscript.
>
> We appreciate your recognition about our method, as well as your constructive advice on some weakness. As a non-native English speaker, we might have got some points that we did not explain clear enough in our manuscript, which could lead to some misunderstandings. For the grammar errors, we are working on correcting and rephrasing, and will get these fixed in the next edition. We would also like to explain about other concerning.
>
> Crystals are known for the ordered structure. Any change in the single site in a crystal can influence the environment of the structure, and consequently influence the general chemical properties. We innovatively designed this work that applies an additional traditional transformer block that calculate attention across the entire crystal graph to overcome the problem of MPNN’s lacking long-range interactions. However, we did not have enough time to further optimize the hyperparameters at the time this manuscript was submitted. We will update some of the results in the next edition, although we believe that there is still a gap between the current result and the potential best result.
>
> For the questions, I would like to answer as follow:
> 1.	Coordination number is aa term we use in chemistry to describe the number of atoms surrounding the central atom within a bonding distance. It can be referred to the number of neighbors of a central node when building crystal graphs.
> 2.	The k-nn graph is a directed graph. In most cases, the edges appear in pairs, however, in some unusual crystals, there are monodirectional edges. This could have strengthened the message passing towards some isolated atoms.
>
> Best regards,
>
> Authors of CrysToGraph

---

> > ### Comment · Reviewer_uVzi · 2023-11-22
> >
> > I thank the authors for their response and clarifying the points about how the K-nn graphs and various chemistry terms. (I also appreciate the discussion of E(3) above for the other reviewer.) However, unfortunately, I have decided to keep my score unchanged for now.
> >
> > I understand that the method is designed to incorporpate scientific knowledge into the ML pipeline. However, the results are presently formulated are not good enough to warrant publication in ICLR in my opinion. I think it would be advisable to further tune the parameters for future iterations.
> >
> > Additionally, the paper is hard to understand for people without a crystallography background. I would recommend either a) rewriting the exposition to be more accessible to the ICLR audience or b) resubmitting the paper to a venue that is a better match. (I am in general supportive of sending hard-science papers to ICLR and similar venues, but the writing should be better tailored to the audience.)

---

### Official Review · Reviewer_ivKJ · 2023-11-02

**Soundness:** 2 fair
**Presentation:** 1 poor
**Contribution:** 2 fair
**Rating:** 3
**Confidence:** 3

**Summary:**

In this paper, the authors mainly tackle that GNNs are limited to capturing long-range interactions. To overcome this limitation, the authors propose a model architecture that can capture the local information with GNN and long-range interactions with a graph-wise transformer. Specifically, two building blocks are proposed: edge-engaged transformer graph convolution (eTGC) and graph-wise transformer (GwT). eTGC is a GNN-based architecture that applies the softmax attention mechanism while the message passing, and GwT is a transformer that is performed on the node embeddings. The authors evaluate their methods for predicting chemical properties on the periodic molecular dataset.

**Strengths:**

* Modeling the periodic molecules is an important subject for material discovery.
* The experiments are fairly conducted on a common benchmark for materials.

**Weaknesses:**

* It is not described why modeling the long-range interaction is the key to modeling the periodic molecules. To my understanding, the tackled GNN's limitation is a general problem in the graph field and does not need to be restricted in modeling the periodic structures.
* From the results of Figure 5 and 6, the effectiveness of modeling the long-range interaction by using the transformer seems to be insignificant, even though the long-range interaction is the main point of this paper. In other words, if the number of eTGC blocks becomes larger, the effect of GwT seems to be weakened and the performances are similar.
* Even though the proposed method leverages the transformers which require more memory space, the performances are not significant compared to the baselines.
* It is not explained what the tasks are. The material properties may not be familiar to the AI researchers and noting the tasks as abbreviations without any explanation could confuse the readers.

**Questions:**

* If many GwT blocks make the local information from the eTGC lost, have you ever applied the eTGC and GwT by crossovering them? (for example, $\text{GwT}(\text{eTGC}(\mathcal{G}, L(\mathcal{G})) \times N$)
* Why do you use softmax attention in eTGC?
* Why do you use the simple transformer architecture for GwT not the graph transformer architectures?
* Why is it insufficient to use deeper GNN layers instead of using transformers?

---

> ### Author Response · Authors · 2023-11-22
>
> Dear reviewer,
>
> Thank you for your time on reviewing our manuscript. We appreciate your advice and questions. As a non-native English speaker, we might have got some points that we did not explain clear enough in our manuscript, which could lead to some misunderstandings. We would like to clarify and response to the comments as follow:
> 1.	It is true that the tackled GNN’s problem is a general problem in the graph field. However, from another aspect, periodicity is not the only character of crystals. Chemical properties and activities of crystals are often strongly influenced by long-range interactions such as long-range symmetry and stacking patterns, sometimes even more than the ionic bonds in short-range [1, 2]. Unlike those dissociated small molecules where long-range interactions like coulomb forces and van der Waals forces can be calculated with equations, the long-range interaction in crystals spans a way larger range and much more complexity to model. From this point, crystallography system is a good demonstration of the tackled GNN’s problem.
> 2.	As in the paragraph above Figure 6, the crystal cells are much smaller in dataset log_gvrh than in mp_e_form. This makes long-range interactions not able to be long enough. However, the GwT is essential in modelling large crystal structures.
> 3.	Although the transformer-based structure requires a relatively large memory space, the baseline which take direct graph, line graph and dihedral graph as input requires more.
> 4.	The target of the tasks will be explained in the appendix in the next edition. Thank you for this advice.
>
> For the questions, we would like answer as follow:
> 1.	Following your suggestion, we tried this type of structure, however, found it not as competible as the original one. Performance of GwT(eTGC(G, L(G)) x 3) x 3 was not significantly different from GwT(eTGC(G, L(G)) x 3) x 1.
> 2.	The major incentive of using softmax attention is to normalize the total message passed to a single node.
> 3.	I use the traditional transformer to take non-ordered datasets. I use the simple transformer to calculate attention across the long-range interactions.
> 4.	Similar with comment 1, we sometimes need to construct the graph from very large structures that contain up to hundreds of atoms instead of a small repetitive unit. Thus, the depth of GNN is never deep enough. In Figure 5 (left), the average of atoms in a crystal cell is 29, and the transformer is not likely able to be substituted by deeper GNN layers.
>
> Best regards,
>
> Authors of CrysToGraph
>
> 1. Cao, Y., Fatemi, V., Fang, S. et al. Unconventional superconductivity in magic-angle graphene superlattices. Nature 556, 43–50 (2018).
> 2. Chen, Y., Lai, Z., Zhang, X. et al. Phase engineering of nanomaterials. Nat Rev Chem 4, 243–256 (2020).

---

### Official Review · Reviewer_cTjz · 2023-11-10

**Soundness:** 1 poor
**Presentation:** 2 fair
**Contribution:** 2 fair
**Rating:** 1
**Confidence:** 5

**Summary:**

This paper proposes a method for predicting chemical properties of materials. Building on the results of previous work, the method utilizes the line-graph of the crystal graph for message passing. For this, the attention mechanism of the transformer architecture is adapted to the case of 3 nodes of the original graph. These 3 nodes are adjacent nodes in the line graph. Additionally, an additional transformer layer that performs message passing on the complete graph is added. For this module a positional encoding is introduced, which merges 4 different positional encodings.

**Strengths:**

The specifically tailored development of machine learning methods for crystal structures is an important problem with relevance for the society. Because of this also machine learning conferences like ICLR should be open for publications that develop these specific methods. However, this publication has certain weaknesses, please see below.

**Weaknesses:**

*Positional Encoding*

The positional encoding is not E(3) invariant and thus is not suitable for modeling the properties of materials. Nodes of the crystal graph are represented with a 3D coordinate of the particular atom in the crystal grid. For a machine learning model to successfully predict the material properties, the feature representation needs to be equivariant or invariant towards E(3) transformations of these coordinates. The proposed positional encoding is not invariant or equivariant to these transformations:

1. Cartesian coordinates: If I understand correctly, then these are extrinsic coordinates. Any function that directly depends on these extrinsic coordinates is not equivariant to E(3) transformations.

2. Fractional coordinates: Are these intrinsic or extrinsic coordinates? I must assume that they are extrinsic, thus the same problems exist as described in 1.

3. and 4. Laplacian embedding and Random walk encoding: These might be useful as they are intrinsic, however Laplacian and Random walk are closely related: There is even a Laplacian associated to a Random walk. Therefore this seems redundant, at least terminology-wise the relationship of the two should be pointed out.


*Multi Head Neighbor Attention*

Multi Head Neighbor Attention appears to be just normal multi head attention on the line graph and should be stated as such and not be presented as a new method.


*Technical remarks and writing style:*

"holistic representation of spatial information" seems to be an awkward terminology. More important would be that this representation results in E(3) equivariance which is a mathematical term which "holistic" is not and thus should be avoided.

Introduction, 2nd paragraph, typo: enxtends

**Questions:**

Which of the positional encodings are intrinsic and which are extrinsic?

**Details Of Ethics Concerns:**

No concerns.

---

> ### Author Response · Authors · 2023-11-22
>
> Dear reviewer,
>
>
>
> Thank you for your time on reviewing our manuscript. I appreciate your recognition of the importance of this topic. However, there are certain scientific flaws in your comment. You may be an expert in machine learning, but clearly you have LITTLE understanding in crystallography.
>
>
>
> E(3) invariance is crucial in modelling dissociated small molecules, however, this does NOT apply to crystal materials. Crystal cells are NOT applicable for the rotations due to the certain construction rules. a, b and c axis are NOT exchangeable. Also, restricted by the PBC nature, crystal cells can expand in the space ONLY in a certain direction, especially in hexagonal, trigonal, monoclinic and triclinic crystals.
>
>
>
> For other comments, I response as follow:
>
> 1. Multi head neighbor attention is NOT just normal multi head attention. As I described in section 3.3.1, The k and v are summed with information from the central node, neighboring node and the edge. Nevertheless, the word neighbor appears mainly to discriminate attention calculated on neighbors and self attention which later applied in GwT.
>
> 2. Thank you for making some contribution to the revision by locating my typo.
>
> 3. Cartesian coordinates and fractional coordinates are extrinsic, Laplacian encoding and random walk encoding are intrinsic.
>
>
>
> Best,
>
>
>
> Author of CrysToGraph

---

> ### Comment · Reviewer_cTjz · 2023-11-22
> **Post-rebuttal: Official Comment by Reviewer cTjz**
>
> Given above rebuttal and the other reviews I do not see a reason to change my review score.
>
> Regarding the authors objection that E(3) invariance would not be needed: a regular crystal lattice has the group of discrete rotations and discrete translations as its symmetry group. However, the proposed extrinsic positional encoding does not result in an equivariance or invariance towards these transformations.
>
> To the authors: It is not my task to make a contribution to your paper but to give a good assessment of its quality and fit to be published at ICLR. I would advise the authors to avoid using all caps in their rebuttal, not to try to undermine the credibility of the reviewer, and to avoid provoking statements (e.g. your point 2 above). I find your wording inadequate for a constructive review process. Further, I have informed the area chair about my concerns.

---

> ### Comment · Reviewer_cTjz · 2023-11-22
> **Official Comment by Reviewer cTjz: References that mention the importance of equivariance / invariance also for crystal graphs:**
>
> References that mention the importance of E(3) equivariance / invariance for crystal graphs:
>
> [1] Yan et al., Periodic Graph Transformers for Crystal Material Property Prediction, Neurips 2022:
> "In addition to being E(3) invariant, periodic graph representations need to be periodic invariant."
>
> [2] Kaba et al., Equivariant Networks for Crystal Structures, Neurips 2022:
> "By using the crystal structure, our approach amounts to defining a group-equivariant convolution kernel on the crystal ..."
>
> [3] Cheng et al., A geometric-information-enhanced crystal graph network for predicting properties of materials, Nature communications materials 2021
> "In this work, we propose a GNN model to accurately predict properties for any crystalline materials, which is invariant to global 3D rotations, translations, and node permutations."
>
> The last reference explicitly also uses the a, b, and c axis parameterization (see Fig. 1 in [3])  of the crystal cell that the authors refer to. Yet, rotation and translation invariant features are required.
>
> The positional encoding proposed here does not have this invariance, not even for the crystal grid with its restricted degrees of freedom. Therefore the proposed positional encoding is not suitable for material property prediction of crystals.

---

### Meta-Review · Area_Chair_RQT7 · 2023-12-09

**Metareview:**

This paper studies crystal material property prediction using transformers. After rebuttals, some technical issues remain, including equivariance of the proposed approach. Thus a reject is recommended.

**Justification For Why Not Higher Score:**

After rebuttals, some technical issues remain, including equivariance of the proposed approach.

**Justification For Why Not Lower Score:**

NA

---

### Decision · Program_Chairs · 2024-01-16

Reject